# Molecular Iodine Has Extrathyroidal Effects as an Antioxidant, Differentiator, and Immunomodulator

**DOI:** 10.3390/ijms22031228

**Published:** 2021-01-27

**Authors:** Carmen Aceves, Irasema Mendieta, Brenda Anguiano, Evangelina Delgado-González

**Affiliations:** Instituto de Neurobiología, UNAM-Juriquilla, Boulevard Juriquilla 3001, Juriquilla, Querétaro 76230, Mexico; aliciairasema@hotmail.com (I.M.); anguianoo@unam.mx (B.A.); edelgado@comunidad.unam.mx (E.D.-G.)

**Keywords:** molecular iodine, antioxidant, differentiator, immune modulator, cancer, mitochondria, peroxisome proliferator-activated receptor (ppar)

## Abstract

Most investigations of iodine metabolism in humans and animals have focused on its role in thyroid function. However, considerable evidence indicates that iodine could also be implicated in the physiopathology of other organs. We review the literature that shows that molecular iodine (I_2_) exerts multiple and complex actions on the organs that capture it, not including its effects as part of thyroid hormones. This chemical form of iodine is internalized by a facilitated diffusion system that is evolutionary conserved, and its effects appear to be mediated by a variety of mechanisms and pathways. As an oxidized component, it directly neutralizes free radicals, induces the expression of type II antioxidant enzymes, or inactivates proinflammatory pathways. In neoplastic cells, I_2_ generates iodolipids with nuclear actions that include the activation of apoptotic pathways and the inhibition of markers related to stem cell maintenance, chemoresistance, and survival. Recently, I2 has been postulated as an immune modulator that depending on the cellular context, can function as an inhibitor or activator of immune responses. We propose that the intake of molecular iodine is increased in adults to at least 1 mg/day in specific pathologies to obtain the potential extrathyroid benefits described in this review.

## 1. Introduction

Iodine in its different chemical forms is captured and used by practically all living beings and is considered a micronutrient in chordates. In vertebrates, iodine is a component of thyroid hormones that is essential for the proper development and functioning of several organs, primarily in the nervous system [1]. However, a significant amount of iodine in the body is non-hormonal and is concentrated in extra-thyroid tissues, where its biological function is barely understood [2]. Several groups have postulated that iodine may have an ancestral antioxidant function in all the cells that concentrate it, from primitive algae to the most recent vertebrates [3,4]. In these cells, oxidized iodine can act as an electron donor neutralizing reactive oxygen species (ROS) or attach to the double bonds of some polyunsaturated fatty acids in cell membranes, making them less reactive to ROS [5]. In addition, it has been shown that iodine binds to lipids, such as arachidonic acid (AA), exerts apoptosis, and/or has differentiation effects on diverse epithelial cells [6,7,8]. Moreover, iodine is uptake and metabolized by immune cells, and depending on the physiological context, this halogen can act as an anti-inflammatory or proinflammatory agent [9,10]. The distribution and action of iodine in organisms depend on the chemical form of iodine that is ingested. For example, molecular iodine (I_2_) is not reduced to iodide (I^−^) in the blood before being absorbed in the gastrointestinal tract [11], induces differential effects [12], and its capture is 40% lower in the thyroid [13]. In fact, under conditions of iodine deficiency, I^−^ appears to be more efficient than I_2_ in restoring the thyroid gland to normality in goiter stages, while I_2_ is more effective in decreasing mammary alterations secondary to iodine deficiency [14]. This article reviews different reports on the effects of iodine as an antioxidant, differentiator and immunomodulator, and does do not include the actions of thyroid hormones.

### 1.1. Safety Concentration

Iodine is a structural component of thyroid hormones, which are essential for differentiation of the nervous system during development and crucial regulators of energetic metabolism. Public health policies have been established to guarantee that populations consume the required amount of iodine to eradicate iodine deficiency disorders. According to the International Council for Control of Iodine Deficiency Disorders (ICCIDD), the recommended dietary allowance of iodine is 150–299 μg/day for normal thyroid functioning, and the maximum limit of iodine intake with the lowest observed adverse effect level (LOAEL) is 1700–1800 μg/day [15,16]. In 1988, the joint of Food and Agriculture Organization of the United Nations and WHO Expert Committee on Food Additives suggested the maximal upper level from all iodine sources of 1 mg/day would be safe for most of the population except those with iodine sensitivity or underlying thyroid disorders. The increased intake of iodide can also have interactions with medications such as lithium or sulfisoxazole [17], but similar studies with molecular iodine do not exist; see Table 1 [15,16]. However, several studies report that iodine supplements at moderately high concentration are well tolerated in euthyroid subjects and that only high doses (>30 mg/day), mainly as I^−^, generate hypothyroidism and goiter, which rapidly revert to normal when these individuals stop taking iodine at high concentrations, see Table 2 [15,18]. Other studies indicate that iodine per se participates in the physiopathology of various organs that uptake it, mainly the thyroid, mammary, prostate, pancreas, and ovaries, and potentially in the gastric, immune, and nervous systems [6]. Moreover, in its molecular form, iodine acts as an antioxidant throughout the body if ingested at concentrations higher than 1 mg/day [19,20]. Dose-response studies in humans have demonstrated that I_2_ at concentrations of 1 to 6 mg/day exhibited significant beneficial actions in benign pathologies like fibrocystic breast disease [21,22], prostatic hyperplasia [23] and polycystic ovaries (unpublished results). The treatments in these studies lasted from five weeks up to two years and did not have any side effects at these concentrations. Some of the dose-response studies also analyzed the highest concentration of iodine (9 and 12 mg/day) and showed the same benefits but accompanied, in some cases, by transient hypothyroidism and/or minor side effects like headache, sinusitis, acne or diarrhea. These effects disappeared when the high dose of supplemental iodine was suspended [24]. Antineoplastic action of the I_2_ supplement without harmful effects on the thyroid has also been observed in mammary and prostatic pathologies in preclinical (rodents and canines) and clinical protocols [25,26,27,28]. Although the thyroid captures 40% less I_2_ than I^−^, the acceptable upper limits for iodine intake during pregnancy are not well defined, and the consequences of excess iodine in newborns are not well documented [15], so the iodine intake in any of its forms above the upper limits is not recommended in pregnant women or infants.

### 1.2. Iodine in Normal Tissues

Although the main uptake of iodine takes place in the thyroid, many other organs take it up (Figure 1), including the salivary glands, gastric mucosa, lactating mammary gland, nervous system, choroid plexus, ciliary body of the eye, lacrimal gland, thymus, skin, placenta, ovary, uterus, prostate, and pancreas, and they can maintain or lose this ability in pathological conditions [1]. The I^−^ transport system in many of these extrathyroidal tissues involves the expression of the sodium iodide symporter (NIS) and/or the anion exchanger Pendrin (PDS/SLC26A4). Recent studies have also demonstrated the direct participation of other transporters including anoctamin 1 (ANO1), cystic fibrosis transmembrane conductance regulator (CFTR) and sodium multivitamin transporter (SMVT) that are capable to take up I^−^ [1]. On the other hand, various studies have shown that I_2_ is captured by an independent mechanism of NIS, PDS, Na^+^ and ATP, but it is saturable and depends on protein synthesis, suggesting a facilitated diffusion system [34]. This mechanism is similar to the one described in marine algae [35], indicating that I_2_ absorption could be an evolutionary conserved system. Indeed, we demonstrated that the thyroid, mammary gland, and prostate can accumulate both types of iodine, which are captured by different mechanisms. The thyroid, lactating mammary gland, and prostate exhibit a significant uptake of I^−^, which is internalized by NIS (inhibited by KClO_4_). Molecular iodine is taken up by these tissues, but also by others like the nubile mammary gland, and NIS does not participate in its internalization [36]. These findings agree with the notion that I_2_ contributes to maintaining the integrity of these organs. Iodine deficiency in rats is accompanied by ductal hyperplasia and perilobular fibrosis in the virgin mammary glands, and the supplement of I_2_ but not I^−^ reverts these alterations [14]. Similarly, the supplement of I_2_ (3–6 mg/day) in patients with fibrocystic breast disease is accompanied by remission of symptoms, as well as significant anti-inflammatory effects [21,22]. Our group has found similar benefits in benign prostatic hyperplasia (BPH) in preclinical and clinical models [36]. In human patients with early BPH (Grade I and II), the supplement of 5 mg/day of Lugol’s solution (mix 1:3; I_2_:KI) for 8 months decreased the prostate-specific antigen (PSA) circulating levels and improved the urinary flow and symptoms scale [23]. These studies agree with epidemiological data that associate the low incidence of breast and prostate pathologies with the moderately high dietary intake of iodine in Asian countries [3,4,37]. These populations consume marine algae daily in their diet, which contain high amounts of iodine in various chemical forms such as I^−^, I_2_ and iodate (IO_3_)^−^. The average consumption of iodine in the Japanese population is 1200 to 5280 µg/day compared to 166 and 209 µg/day in the United Kingdom and the United States, respectively [33,38]. However, despite the high nutritional intake of iodine, Asia does not differ from the rest of the world in the prevalence of thyroid disorders [37].

### 1.3. Antioxidant Effects

Iodine is considered an ancestral antioxidant and its action is conserved throughout phylogeny [2]. Laminaria brown algae contain an iodine concentration 300,000 times higher than any other living organism, and inorganic iodine acts as a scavenger of various reactive oxygen species (ROS) [35,39]. Similar antioxidant effects have been described in other photosynthetic organisms, as well as in some invertebrates such as polyps of the jellyfish *Aurelia aurita* and urchin larvae [40,41]. In vertebrates, micromolar amounts of iodine decrease damage by ROS, increasing the total antioxidant status in rat and human serum [20] and preventing lipid peroxidation in the eyes of rabbits [42] and in several tissues of vertebrates [43,44,45]. The iodine released by deiodination of thyroxine has been shown to be an antioxidant agent and an inhibitor of lipoperoxidation [43]. Molecular iodine supplements decrease lipid peroxidation in normal and tumor mammary tissues from rats with methyl nitrosourea (MNU)-induced mammary cancer [13] and prevent the cardiac damage induced by the antineoplastic agent doxorubicin when I_2_ (0.05% in drinking water) is administered 2 days before starting the antineoplastic treatment [19]. Moderate iodine diets improve the lipid profile in mice, increasing low density lipoprotein receptors and scavenger receptor class B type 1 (SR-B1) in liver [46]. Moreover, iodine supplementation decreased hypercholesterolemia in overweight women [47]. More recently, our group showed that a moderate I_2_ supplement prevented the pancreatic damage secondary to hypothyroidism by methimazole, normalizing thyroid hormone synthesis in the thyroid and preventing the oxidative status in pancreatic tissue [48]. Several studies suggest that iodine works by neutralizing ROS, or by acting as a free radical iodinating tyrosine, histidine, and double bonds of polyunsaturated fatty acids in cell membranes, making them less reactive to ROS [4,49]. However, the antioxidant effect of iodine could be more complex and include various mechanisms (Figure 2). In a model of prostatic hyperplasia, our group demonstrated that I_2_ supplements prevent testosterone-induced oxidative stress, decreasing lipoperoxidation but also inhibiting the activity of both nitric oxide synthase (NOS) and type 2 cyclooxygenase (Cox2). The I_2_ supplement also inhibit the formation of prostaglandins with equivalent intensity to that observed with Celecoxib (a specific Cox2 inhibitor). The effect on Cox2 inhibition can occur by deactivating the heme iron active site or as a competitor of its main substrate, arachidonic acid (AA). In the latter case, the formation of 6-iodolactone (6-IL) from AA can decrease the formation of prostaglandins, or 6-IL acts as a direct inhibitor of the enzyme [50]. Another recent proposal is the interaction of I_2_ with the nuclear factor erythroid-2-related factor-2 (Nrf2) pathway [51]. Nrf2 is a promoter of the antioxidant response to endogenous and exogenous stressors that trigger the expression of phase II protective antioxidant enzymes such as superoxide dismutase (SOD) and catalase (Cat) [52]. Under basal conditions, Nrf2 is anchored to the cytoplasm through the actin cytoskeleton-binding protein 1 (Keap1). Iodination of Keap1 results in the release and translocation of Nrf2 to the nucleus. After Nrf2 heterodimerizes with small Maf proteins and binds to the antioxidant response element (ARE), SOD and Cat become overexpressed [51].

### 1.4. Antiproliferative and Apoptotic Actions

Since the 1940s it has been known that iodine, in addition to be a structural part of thyroid hormones, also participates in the function and proliferation of thyroid cells. An excess of I^−^ causes inhibitory actions that include decreased iodine organification and hormonal secretion, thyroglobulin proteolysis, decreased glucose and amino acid transport, protein and RNA biosynthesis, and significant inhibition of thyrocyte proliferation under both in vitro and in vivo conditions [49]. The specific mechanism by which iodine performs all these modifications has not been fully elucidated, but a multifaceted mechanism has been postulated and includes the participation of transforming growth factor beta-1 (TGF-β1), triiodothyronine (T3) and iodolipids such as 6-IL or 2-iodohexadecanal (2-IHDA). Moreover, all the inhibitory actions of iodine can be reversed with drugs that block the enzyme thyroid peroxidase (TPO), such as methylmercaptoimidazole (MMI) or propylthiouracil (PTU) [49]. In the presence of H_2_O_2_, TPO oxidizes I^−^ and covalently binds it to proteins or lipids. The specific species of iodine generated by TPO has not been identified, but there are several candidates, such as I^−^, I0 (free radical of iodine), IO^−^ (hypoiodite) and I_2_ [49]. Vitale et al. [53] showed that an excess of KI (10–50 mM) induces apoptosis in primary thyrocyte cells, but if TPO activity is blocked with PTU, the apoptotic effect of I^−^ is eliminated. In addition, lung cancer cells (without absorption of natural iodide) transfected with NIS or NIS/TPO, a supplement of KI (30 mM), induced apoptosis only in cells transfected with NIS/TPO, indicating that oxidation of I^−^ by TPO is required to exert apoptotic effects [54].

In terms of carcinogenesis, the overproduction of ROS, such as single oxygen (O_2_), superoxide anions (O_2_^−^), hydrogen peroxide (H_2_O_2_) and hydroxyl radicals (°OH), is a hallmark related to the etiology and progression of cancer [55]. ROS have a wide range of cellular and molecular effects resulting in mutagenicity, cytotoxicity, and changes in gene expression. The notion that I_2_ is the chemical form responsible for antineoplastic effects originates from the first descriptions of the consumption of seaweed or Lugol’s solution [4]. Previously, we mentioned that seaweeds contain iodine in several chemical forms although the exact proportion is not known [33]. Traditional Eastern breast cancer medicine has long used iodine-rich seaweeds as a cancer treatment to “soften” tumors and “reduce” nodulation [56]. The addition of small proportions (2 to 5%) of *Laminaria angustata*, *Porphyra tenera* or *Laminaria religiosa* to the diet significantly delays the occurrence of tumors in rats treated with the chemical carcinogen, 7,12-dimethylbenzanthracene (DMBA) [57,58]. The first report demonstrating that iodine exhibited an antineoplastic effect in extrathyroidal tissues was in the rat mammary cancer model induced by DMBA, using 0.05% Lugol’s solution [59]. Later, our group reported that in this model, KI, I_2_, or Lugol’s solution can induce antineoplastic actions. The protective effect of 0.1% KI is lost when the enzyme lactoperoxidase (LPO), which is present in mammary cancers, is inhibited by MMI, indicating that I^−^ from KI needs to be oxidized to have the apoptotic effect [60]. In this study, our group also demonstrated that I_2_ prevents DMBA-induced DNA adduct formation in pre-malignant and cancer tissues. This finding is particularly relevant since LPO can oxidize natural or synthetic estrogens to catechol estrogens [61]. The resulting estrogenic quinones have been shown to react with DNA to form mutagenic adducts that can initiate or promote cancer [62]. This notion agrees with the report of Cavalieri’s group showing that higher levels of E2-DNA adducts are present in the urine of breast cancer patients and women at high risk for this disease [63].

Various groups have described apoptosis effects of iodine in several cancer cell lines and proposed different mechanisms and pathways (Figure 3). The most studied effects include a direct action, where the oxidized iodine dissipates the mitochondrial membrane potential, thereby triggering mitochondrion-mediated apoptosis [64], and an indirect effect through iodolipids formation and the activation of peroxisome proliferator-activated receptors type gamma (PPARγ) [65].

It is well known that the mitochondrial membrane potential (MMP) is required for a variety of mitochondrial functions including protein import, ATP production, and regulation of metabolite transport. The mitochondrial intermembrane space contains proteins that can induce apoptosis involving caspases (e.g., cytochrome c) or execute a caspase-independent apoptotic death program through the apoptosis-inducing factor (AIF) or through the release and degradation of the antiapoptotic protein Survivin (SVV). The release of these factors requires abatement of the MMP, and thiol depletion is a powerful trigger [66,67]. Molecular iodine treatment is accompanied by depletion of cellular thiol content and dissipation of the MMP in estrogen-responsive (MCF-7) and non-responsive (MDA-MB-231) human cell lines. In addition, the pre-incubation of MCF-7 cells with N-acetylcysteine (NAC), a thiol-containing agent, prevents the apoptotic effect of I_2_ [64]. Comparative studies of mitochondria isolated from tumoral (TT) versus extra-tumoral (ET) human breast tissue showed that the I_2_ treatment increased mitochondrial permeability in TT and decreased it in ET, suggesting a differential sensitivity toward iodine in both physiological conditions [68].

The indirect action of I_2_ could be exerted by the formation 6-IL previously detected in thyroid tissue of rat, pig, horse, and human [69,70]. Although the specific iodinated components have not yet been characterized in other tissues, several studies have reported elevated prostaglandin levels in cancerous tissues compared to normal tissues [71]. Prostaglandins are produced from AA by Cox2, indicating high levels of AA in several tumors [72]. In relation to the mammary gland, we reported that MNU-induced tumors contain four times higher basal concentrations of AA, and after 0.05% I_2_ treatment, 6-IL levels were 15-fold higher than in normal mammary tissue, suggesting a role for 6-IL in the antiproliferative effect of I_2_ [65]. These findings have been corroborated in human cancer cell lines where lipids like 6-IL were identified after I_2_ treatment [73] or where the addition of I_2_ or 6-IL triggered apoptosis [74,75]. In this regard, the consistent observations that cancer cells are more sensitive to I_2_ than normal cells [73,74,75] led us to propose that the high concentration of AA in tumoral cells is the crucial component that, when iodinated, is responsible for the antiproliferative effect of I_2_ [65].

In the search for cellular mechanisms associated with iodine effects, studies from our laboratory demonstrated that both I_2_ and 6-IL supplementation significantly modified the expression of PPARs [76]. These receptors, originally associated with lipid metabolism regulation, are widely expressed and form part of the nuclear receptor family that binds thyroid hormones, steroids, and vitamins. To date, three isotypes called PPARα, PPARβ/δ, and PPARγ have been identified. These three subtypes display differential tissue distribution, and each is involved in specific functions such as early development, cell proliferation, differentiation, apoptosis, and metabolic homeostasis [77]. In our experiments, 20–200 μM I_2_ increased the expression of PPARγ mRNA and protein, decreased the expression of mRNA for PPARα, and had no effect on PPARβ/δ expression in MCF-7 cells. We also showed that 6-IL is a specific agonist of PPARγ with an in vitro affinity 6 times higher than AA [76]. These findings agree with the observation that the affinity and selectivity of the PPARγ isoform for some fatty acids is increased by the conformational changes resulting from the incorporation of halogens (phenyl acetate < phenyl butyrate < p-chlorophenyl acetate < p-iodophenyl butyrate) [78]. Moreover, recent reports have shown that antineoplastic effects of iodine or iodolipids are exerted on different types of cells that can take up I_2_ and exhibit apoptotic induction by PPARγ agonists. Such cells include prostate, lung carcinoma, pancreas carcinoma, melanoma, glioblastoma, and neuroblastoma cells [79].

### 1.5. Effects on Cellular Differentiation

Another possible effect of iodine is the induction of cellular differentiation. Iodine plays a central role in thyroid physiology by maintaining the integrity of thyroid epithelium [23]. Epidemiological studies associating iodine intake and thyroid cancer have led to controversy. Some suggested that chronic iodine deficiency is firmly established as a risk factor for follicular thyroid cancer, whereas others suggested that iodine supplementation programs could increase the incidence of papillary thyroid cancer in chronic iodine-deficient populations [80]. In relation to differentiation induction, cancer studies have shown that moderate iodine supplements prevent the transformation from differentiated to anaplastic thyroid cancer, the most aggressive type of thyroid cancer with a median survival of 4–12 months from the time of diagnosis [81,82]. In vivo studies of mammary cancer (MNU-induced model, xenografts in the nu/nu Foxn1 mouse, and canine and human patients) have shown that I_2_ treatment increases the expression of NIS, PDS, LPO, and/or estrogen alpha receptors (ERα) and reduces the invasive and metastatic inducers like vascular endothelial growth factor (VEGF), urokinase-type plasminogen activator (uPA), Bcl2, and SVV, indicating a consistent effect on differentiation [25,26,28,83]. Similar results were obtained in trophoblast cells exposed to iodine supplementation increasing the synthesis of chorionic gonadotropin [84]. Moreover, studies related to stemness markers in breast and cervical cancer cells indicated that I_2_ supplementation decreased the stem cell-like population, transforming the survival cells into non-invasive type cells incapable of generating xenografts [85,86]. In both conditions PPARγ receptors were increased.

### 1.6. Immune Modulator

It is widely accepted that iodine exerts important actions on the immune system [87]. The thymus, as well as many immune cells, have the ability to capture and metabolize iodine [88,89]. This element can act as an inhibitor or activator of the immune response depending on the cellular context (Figure 4). In cell damage models induced by non-infectious agents it has been shown that iodine prevents the inflammatory response with a radical scavenging effect. It has described that iodine promoted inhibition of ROS production and oxygen consumption of human polymorphonuclear leukocytes [90], decreased neutrophil chemotaxis [91] and inhibited human complement, mast cell degranulation, nitric oxide and TNF-α production by murine and human macrophages [9,92]. Iodine is oxidized by myeloperoxidase into free radical iodine and is used to kill bacteria [93]. This ancestral defense mechanism has been documented in *Roseolovirus* spp. bacteria as a defense against other species [94]. In addition, iodide supplementation increases granulocyte chemotaxis to inflamed areas and improves phagocytosis of bacteria [95]. Besides, under adaptive immune response, micromolar concentrations of iodine improve the Th2 response of leukocytes from normal subjects preactivated with PMA, increasing the release of IL6, IL8-CXCL8 and IL10 [9]. Marani and Venturi described that in the central region of Italy, partial iodine deficiency (urinary excretion of 4 mg/day) is enough to maintain euthyroid conditions, but it is accompanied by serious alterations in the immune response of school-age children. Lugol’s oral solution supplement (2 mg/week) for 8 months restored the normal immune response evaluated by skin tests [96]. Similar responses to Lugol’s supplement were observed in patients with infections in which the immune system was compromised, such as granulomatous, lepromatous, syphilitic, and fungal tuberculous lesions [97].

Regarding the participation of iodine in the immune response against intracellular antigens (virus or cancer), studies of our group have shown that I_2_ supplementation, alone or in co-administration with conventional chemotherapies, is accompanied by the activation of anti-tumor immune responses in several models (rodents, dogs and humans). The remaining tumors from these treatments, which contain multiple necrotic areas, also show CD8^+^ immune cell infiltration [13,26,28]. Transcriptomic analysis of human samples showed that I_2_ supplementation up-regulated Th1 and Th17 pathways including differentiation by activation of T-cell receptor, cytotoxicity by NK and CD8^+^, lymphocyte migration and formation of tertiary lymphoid structures [26,98]. In accordance with these data, micromolar supplementation of I_2_ improves the Th1 response in leukocytes from normal subjects by increasing the release of IL2 and IFN-γ cytokines [9].

The signaling pathways underlying these immune responses have not been deeply studied. However, the activation of PPARγ receptors might be involved, since these receptors participate in the modulation of several immune cells [99]; although it is generally accepted that PPARγ exerts anti-inflammatory responses on infections, its participation in the antitumoral immune response has also been suggested. One report using cyclophosphamide metronomic therapy showed the activation of the immune cascade, and PPARγ was proposed as the main inducer [100]. Our group identified a similar activation through breast cancer RNAseq analysis; PPARγ was positively correlated with changes in IRF1, STAT1, and IRF4 [26]. Another possibility currently explored in our laboratory is that I_2_ as an oxidized agent can exert epigenetic modifications associated with the activation of important demethylase enzymes like DMT3 [101], unpublished results). In fact, other natural antioxidants, such as resveratrol or curcumin, exert part of their effects by modifying the methylation/demethylation equilibrium genes for the differentiation of Th1 (T-bet and STAT4) or Th2 (GATA3) [102].

## 2. Discussion/Conclusions

Molecular iodine in vertebrates acts in the following ways:as an ancestral antioxidant by combining or competing with free radicals for membrane lipids, proteins, and DNA, increasing the expression or activity of antioxidant enzymes, or inactivating proinflammatory pathways stabilizing cellular redox status;as an inducer of antiproliferative, differentiation or apoptotic mechanisms by modulating mitochondrial potential or forming iodolipids and activating nuclear receptors;as an immune modulator acting directly on specific immune cells; andas a constituent part of thyroid hormones.

Although iodine excess is generally well tolerated, it may induce physiological changes in susceptible groups, particularly those previously exposed to iodine deficiency, thyroid defunctions, pregnant women, or infants. We propose that molecular iodine intake be increased in adults to at least 1 mg/day in specific pathologies to obtain the potential extrathyroidal benefits described in the present review.

## Figures and Tables

**Figure 1 ijms-22-01228-f001:**
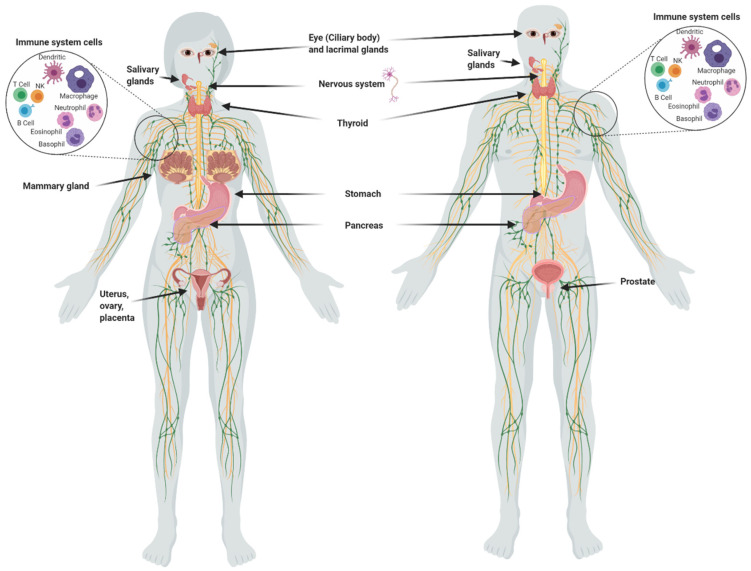
Organs and tissues that take up iodine.

**Figure 2 ijms-22-01228-f002:**
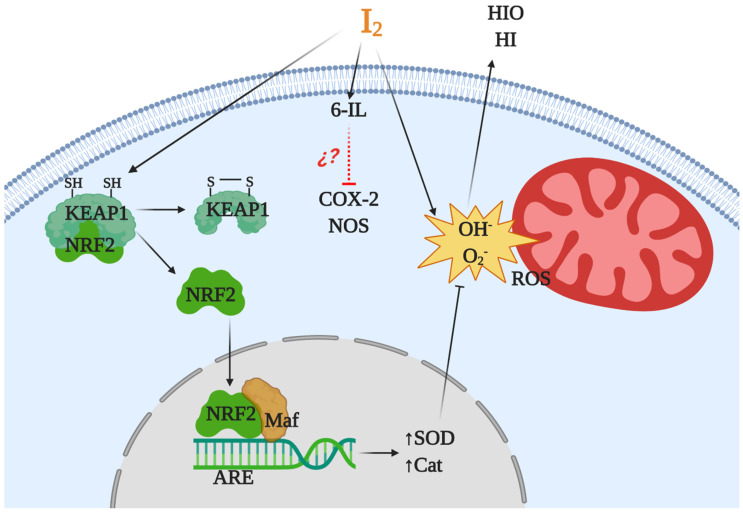
Antioxidant mechanisms of molecular iodine (I_2_). I_2_ acts as a scavenger of a reactive oxygen species (ROS) like hydroxyl radicals (OH) or superoxide anions (O_2_) generating neutral components hypoiodous acid (HIO) or hydroiodic acid (HI). I_2_ in combination with arachidonic acid (AA), and generating the iodolipid 6-iodolactone (6-IL), inhibits the activity of proinflammatory enzymes like nitric oxide synthase (NOS) and cyclooxygenase type 2 (Cox2). In addition, the iodination of the cysteine-rich protein Keap1 releases and promotes the nuclear translocation of nuclear factor erythroid-2-related factor-2 (Nrf2) that with Maf activates the antioxidant response element (ARE), inducing overexpression of antioxidant enzymes type II like superoxide dismutase (SOD) and catalase (Cat).

**Figure 3 ijms-22-01228-f003:**
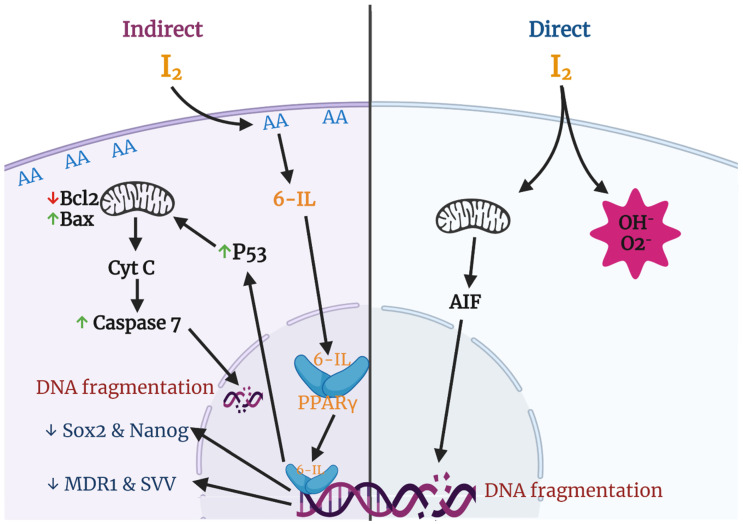
Apoptotic and differentiation mechanisms of molecular iodine (I_2_). In the direct pathway the oxidized iodine dissipates the mitochondrial membrane potential triggering mitochondrion-mediated apoptosis. The indirect pathway includes the iodination of arachidonic acid (AA), generating 6- iodolactone (6-IL), and the activation of peroxisome proliferator-activated receptors type gamma (PPARγ). The activation of PPARγ could induce the p53-caspase apoptotic pathway and/or the inhibition of markers related to stem cell maintenance (Sox2, Nanog), chemoresistance (multidrug resistance protein 1; MDR1), and survival (Survivin; SVV).

**Figure 4 ijms-22-01228-f004:**
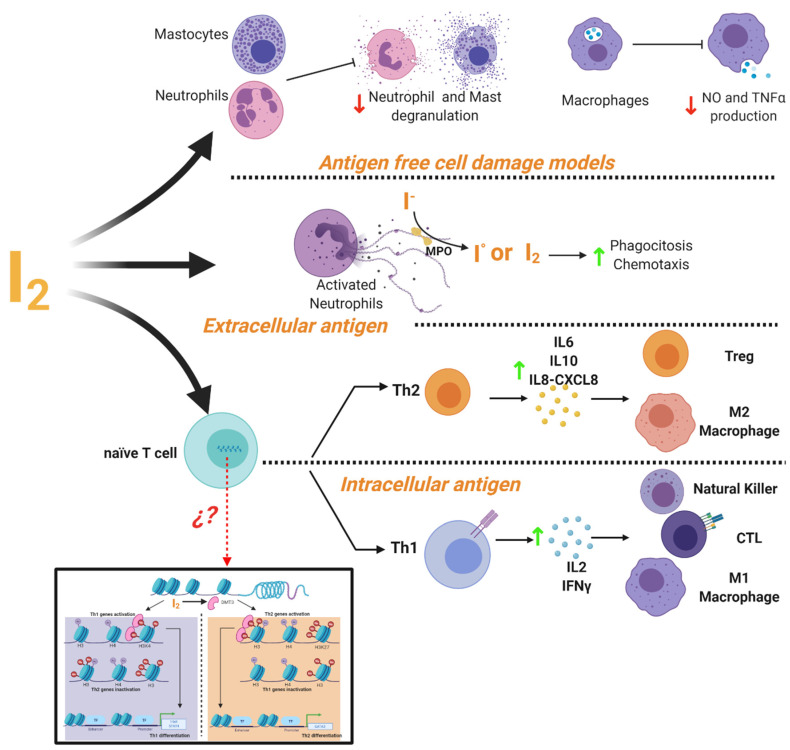
Mechanisms of immune modulation by molecular iodine (I_2_). Evidence showed that I_2_ the mediated activation or inhibition of immune responses depending on the cellular context. In a cell damage environment without antigens iodine supplementation inhibits the degranulation of neutrophil and mast cells, as well as the production of macrophages NO and TNFα. In response to extracellular antigens (e.g., bacteria or fungi), I_2_ improves the phagocytosis and chemotaxis of activated neutrophils. It also activates the Th2 response and the increase of cytokine secretion (IL-6, IL-10, IL-8-CXCL8) promoting the activation and differentiation of Treg lymphocytes and M2 macrophages. In response to intracellular antigens (e.g., virus-infected cells and cancer cells), I_2_ activates the Th1 response and increases the cytokine production of IL-2 and IFNγ, promoting the activation and proliferation of effector cells (NK, CTL and M1 macrophages). This differential response might be associated with epigenetic modifications facilitated by the I_2_-mediated activation of the demethylase enzyme (DMT3) that regulates the Th fate by activating and/or inactivating the specific genes for the differentiation of Th1 (T-bet and STAT4) or Th2 (GATA3).

**Table 1 ijms-22-01228-t001:** Predisposing Risk Factors Associated with permanent Iodine-Induced Thyroid Dysfunction.

Individuals with Underlying Thyroid Disease
Graves’s disease
Hashimoto thyroiditis
Euthyroid with a history of subacute thyroiditis
Euthyroid with a history of postpartum thyroiditis
Euthyroid with a history of type 2 amiodarone–induced thyroiditis
Euthyroid with post-hemithyroidectomy
Euthyroid after interferon-γ therapy
Individuals with a family history of goiter or thyroiditis
Individuals with chronic iodine deficiency
Fetuses, preterm neonates, and newborn infants exposed to high doses of iodine through the placenta and milk
Elderly people with subclinical hypothyroidism
Patients taking medications such as expectorants or amiodarone that contains high concentrations of iodine
Patients with certain nonthyroidal disease such as chronic dialysis and cystic fibrosis, especially those taking sulfisoxazole.
Patients taking lithium

**Table 2 ijms-22-01228-t002:** Sources and Effects of Excess Iodine.

Source of Iodine	Iodine Dose (mg/day)	Treatment Time	Chemical Form of Iodine	Effects on Thyroid Function	Ref
Iodopovidone (5% solution) mouthwash using 2–4 mL	14–28	Days-weeks	I_2_	Values remain within normal range	[17,18,29]
Amiodarone		Months-years	Iodide		[15,30]
1 tablet (100 mg)	3			Thyrotoxicosis (2%) Hyperthyroidism (1%)	
1 tablet (600 mg)	21			Hypothyroidism (2–10%)	
Iodinated contrast medium (200 mL/dose)	7–10	One dose	Iodide	Hyperthyroidism or Hypothyroidism (1–2%)	[15,31]
Seaweed Blended brown seaweed (1 bowl, 250 mL soup)	1–3	Weeks-months	Iodide, I_2_	Normal values or transient subclinical hypothyroidism (2–10%)	[32,33]
High level of consumption (>6 g seaweed/day)	>20			Risk of papillary thyroid cancer (1–10%)	
KI supplements Water solution (5–15 mg)	>2	Days-weeks	Iodide	Transient subclinical hypothyroidism,	[15,17,18]
1 tablet (50 mg)	>30			Thyrotoxicosis (2–10%) TPOAb, TgAb (6–20%)	
Purified water solutions (8 mg/L per tablet) 1 tablet	1–5	Months-years	I_2_	Normal values	[15,17,18]
4 tablets	10–32			Transient hypothyroidism and goiter TPOAb, TgAb (3–16%)	
Aqueous I_2_ solution I_2_ water solution; Lugol’s solution, or (1–2 tablets (3 mg per tablet)	1–6	Months-years	I_2_, I_2_-iodide	Values remain within normal range	[17,18,21,22,26]
3–4 tablets (3 mg per tablet)	9–12			Transient subclinical hypothyroidism, headache, sinusitis, diarrhea acne (6–20%)	
Mix yodica	1–3			Values remain within normal range	

KI, potassium iodide; I_2_, molecular iodine, TgAb, thyroglobulin antibody and TPOAb, thyroid peroxidase antibody positive titer. Modified from [6].

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
