# Peer review of "Molecular Iodine Has Extrathyroidal Effects as an Antioxidant, Differentiator, and Immunomodulator"

_ijms, 2021, doi:10.3390/ijms22031228_

Round 1

Reviewer 1 Report

The Authors, in their review, did not provide convincing arguments for the safety of administration of high doses of molecular iodine. The administration of iodine at a dose of 1 mg/kg in selected diseases is very controversial.

Author Response

The authors, in their review, did not provide convincing arguments for the safety of administration of high doses of molecular iodine. The administration of io dine at dose of 1 mg/kg in selected diseases is very controversial

Clinical studies regarding the intake of molecular iodine are still scarce; but we totally agree with the referee, that 1 mg/kg is excessive and with deleterious effects on the thyroid and may be on other organs.

However, there are at least four long-term clinical studies (from 6 months to 5 years) analyzing the intake of molecular iodine at millimolar quantities (1 to 9 mg/day) that exhibits beneficial effects on breast fibrosis or mastalgia (Ghent 1993, and Kessler 2004, 2009), prostatic hyperplasia (Anguiano 2010) and breast cancer (Moreno-Vega 2019). In all cases, there are no harmful effects on thyroid function or general health in the range of 1 to 6 mg/day. As described in Table 2 of this review, moderate hypothyroidism occurs when more than 6 mg (9 to 13 mg/day) of iodine is intake and this condition is reversed without consequences when the consumption stopped. These studies agree with many publications in preclinical studies (mice, rats, and dogs) where the supplement of molecular iodine even at higher doses does not generate any long-term thyroid dysfunction or other side effects. In 2005, our group showed that molecular iodine uptake by the thyroid is significantly lower (40% less) than iodide. This fact helps us to explain the thyroid tolerance to high concentrations of molecular iodine. In addition, we also demonstrated that molecular iodine is an effective antioxidant in both normal or cancerous tissues (mammary gland and hearth) (García-Solis et al, 2005). We are also aware that many of the iodized solutions contain molecular iodine and potassium iodide. In this regard, Dr. Kessler's study shows that KI3 (potassium iodide / I2) solutions are more stable in serum, and that, even at an excess, there are no side effects (Kessler, 2009).

In fact, in the WHO statement, it is stated that… “International reference values for upper intakes… are based upon an interpretation of three small pharmacokinetic dose−response studies evaluating the effects of subchronic iodine exposures in euthyroid adults.17–19 In these short-term studies, though thyroid function tests generally remained within normal ranges despite exposure to up to 4500 ug/day,18 iodine intakes of >500 ug/day over several weeks can induce subtle, reversible changes in the pituitary−thyroid function in adults, probably by inhibiting TH synthesis and/or release. The lowest observed adverse effect level (LOAEL) proposed for iodine intakes based on these studies is 1700−1800 ug/day, based on a mild increase in thyroid-stimulating hormone (TSH) that was not associated with clinical adverse effects”. … The United States Institute of Medicine used a UF of 1.5, bringing the UL to 1100 ug/day,21 whereas the Euro- pean Union Scientific Committee on Food used a UF of 3 to reach a UL of 600 ug/day,20 due to differences in interpretation of the LOAEL effect.16,22 (reference 35).

For all the above, we are proposing that supplement of 1mg per day (not per Kg) is secure and effective. We consider that the importance of our review is precisely to highlight the differential effects of molecular iodine. We include several of these considerations in the text of the manuscript.

Reviewer 2 Report

The authors, in the review article titled "Molecular iodine. Antioxidant, differentiator, and immune modulator factor”, have discussed and highlighted the biological properties of iodine.

In general, it’s a well written manuscript covering different aspects of molecular iodine. However, few changes are required to complete the manuscript.

  1. The punctuation in the title appears to be odd. Please edit it to reflect the content in the final version of the manuscript.
  2. A section on interactions between iodine and medications will be required.
  3. Is there any effect of age on the actions of iodine? Any particular age group more or less susceptible? 
  4. The goals of this review manuscript have been poorly described in the abstract (page 1, line 10) and instruction (page 2,line 48). These need to be updated.
  5. A section on the toxic effects (not thyrotoxicosis) of molecular iodine needs to be added.

Author Response

1. The punctuation in the title appears to be odd.. Please edit …

We agree with this observation, we changed the title for: Molecular iodine has extrathyroidal effects as an antioxidant, differentiator, and immunomodulator

 2. A section on interactions between iodine and medications will be required.

There are few studies related to iodine supplementation and medications, we only found 2, the interaction of iodide with lithium and sulfisoxazole. We do not found information with molecular iodine. We add a phrase with this information in the new version of the Ms.

 3. Is there any effect of age on the actions of iodine? Any particular age group more or less susceptible?

Yes, the excess iodine intake in pregnant and infants are poorly studied, and perhaps it is not recommended as supplement. We added a phrase with these indications.

4.the goals of this review manuscript have been poorly described in the abstract …These need to be updated

We include a phrase in the abstract with this information.

5. A section on toxic effects (not thyrotoxicosis) of molecular iodine needs to be added.

 As we responded to referee 1, few clinical studies with molecular iodine exist, and side effects only occur when ingested concentrations are higher than 6 mg/day. The non-thyroid effects described with more than 6 mg/day include headache (20%); sinusitis (12%); nausea (9.9%); acne (9.0%); back pain (9.0%); diarrhea (9.0%); dyspepsia (8.1%); rash (8.1%); and abdominal pain (6.3%) and disappears as soon as consumption is withdrawn or when the dose is reduced. On the contrary, in our study in women with breast cancer (Moreno-Vega, 2019), the iodine supplement (3 mg/day), together with chemotherapy, attenuates many of the drugs' side effects, and this seems to be due to it antioxidant effect. We added a phrase with these indications.

Round 2

Reviewer 1 Report

The Authors have considerably improved their manuscript and draw more cautious conclusions from their review. 

Neverthless, some minor corrections are needed:

Page 12

line 417: please add Molecular in the beginnig of this line: "Molecular iodine in vertebrates acts in the following way.."

line 427: please add "molecular" to the sentence beginng with "We propose.." It should be: "We propose that molecular iodine intake be increased in adults.."

Author Response

Neverthless, some minor corrections are needed:

Page 12

line 417: please add Molecular in the beginnig of this line: "Molecular iodine in vertebrates acts in the following way.."

line 427: please add "molecular" to the sentence beginng with "We propose.." It should be: "We propose that molecular iodine intake be increased in adults.."

We agree with the referee and the suggestions have been made.